# Longitudinal stability of medial temporal lobe connectivity is associated with tau-related memory decline

**Quanjing Chen[1,2]\*, Adam Turnbull[1,3], Timothy M Baran[3,4], Feng V Lin[1,2,5,6,7,8]**

[1]Elaine C. Hubbard Center for Nursing Research on Aging, School of Nursing, University of Rochester Medical Center, Rochester, United States; [2]Department of Psychiatry, School of Medicine and Dentistry, University of Rochester Medical Center, Rochester, United States; [3]Department of Imaging Sciences, School of Medicine and Dentistry, University of Rochester Medical Center, Rochester, United States; [4]Department of Biomedical Engineering, University of Rochester, Rochester, United States; [5]Department of Neuroscience, School of Medicine and Dentistry, University of Rochester Medical Center, Rochester, United States; [6]Department of Neurology, School of Medicine and Dentistry, University of Rochester Medical Center, Rochester, United States; [7]Department of Brain and Cognitive Sciences, University of Rochester, Rochester, United States; [8]School of Medicine, Stanford University, Stanford, United States

**Abstract** The relationship between Alzheimer's disease (AD) pathology and cognitive decline is an important topic in the aging research field. Recent studies suggest that memory deficits are more susceptible to phosphorylated tau (Ptau) than amyloid-beta. However, little is known regarding the neurocognitive mechanisms linking Ptau and memory-related decline. Here, we extracted data from Alzheimer's Disease Neuroimaging Initiative (ADNI) participants with cerebrospinal fluid (CSF) Ptau collected at baseline, diffusion tensor imaging measure twice, 2 year apart, and longitudinal memory data over 5 years. We defined three age- and education-matched groups: Ptau negative cognitively unimpaired, Ptau positive cognitively unimpaired, and Ptau positive individuals with mild cognitive impairment. We found the presence of CSF Ptau at baseline was related to a loss of structural stability in medial temporal lobe connectivity in a way that matched proposed disease progression, and this loss of stability in connections known to be important for memory moderated the relationship between Ptau accumulation and memory decline.

\*For correspondence: quanjingchen@gmail.com

## Introduction

Alzheimer's disease (AD) is characterized by the presence of both amyloid-beta (Aβ) and tau proteins that lead to neurodegeneration (*Jack et al., 2015*). The dominant view in the literature is that Aβ pathology triggers tau aggregation, strongly supported by evidence from autosomal dominant AD (*Hardy, 2002*). However, there have been important recent findings from sporadic AD that suggest that hyperphosphorylated tau proteins that are no longer governed by normal cellular removal mechanisms play a necessary and underappreciated role in AD progression (*Musiek and Holtzman, 2015*), with some authors even suggesting that this may be the primary step in the disease needed for subsequent AB aggregation (*Arnsten et al., 2020*; *Braak and Del Tredici, 2015*). In addition, a recent study found that tau-positron emission tomography (PET), but not β-amyloid–PET, could predict longitudinal brain atrophy in AD (*La Joie et al., 2020*). Perhaps for this reason, across animal

and human studies, phosphorylated tau (Ptau), rather than Aβ, seems to better reflect cognitive deficits (*Giannakopoulos et al., 2003*; *Hedden et al., 2013*; *Huber et al., 2018*; *Mitchell et al., 2002*; *Bennett et al., 2004*; *Schöll et al., 2016*). However, for Ptau to become a potentially meaningful therapeutic target for preventing or slowing dementia progression, it is essential to better understand the mechanisms linking Ptau and cognitive deficits.

Both in vivo and in vitro findings support that abnormal tau appears to originate in the anterior medial temporal lobe (MTL) before spreading to anterior-temporal (AT) and posterior-medial (PM) regions of the cerebral cortex (*Braak and Del Tredici, 2015*; *Cho et al., 2016*). Cumulative evidence suggests that AD is a *connectome* disease (*Acosta-Cabronero et al., 2010*, *Dai and He, 2014*, *Damoiseaux et al., 2009*, *Delbeuck et al., 2003*, *Rose et al., 2000*), and abnormal tau spreads via neuronal connections (*Clavaguera et al., 2009*). MTL is strongly connected with both the AT and PM systems, with anterolateral entorhinal cortex showing preferential connectivity with anterior regions overlapping with the AT system and parahippocampal gyrus and posteromedial entorhinal cortex with posterior regions overlapping with the PM system (*Maass et al., 2015*; *Navarro Schröder et al., 2015*). The accumulation of abnormal tau in the MTL and subsequent spread to these cortical memory systems are likely to be particularly relevant for understanding memory deficits in AD, as these systems play dissociable roles in human memory: the AT system is specific for object-related stimuli (i.e., semantic memory) and the PM is specific for remembering scenes (*Davachi et al., 2003*; *Epstein and Kanwisher, 1998*; *Wang et al., 2020*). Taken together, these findings suggest that damage to the integrity of neural connections between the MTL, AT, and PM may be a mechanism by which Ptau accumulation leads to cognitive decline.

Several studies have examined the relationship between the functional (*Berron et al., 2020*; *Cope et al., 2018*; *Franzmeier et al., 2020*; *Franzmeier et al., 2019*; *Maass et al., 2015*; *Ossenkoppele et al., 2019*) and structural connectivity and tau pathology (*Jacobs et al., 2018*; *Kim et al., 2019*). Tau propagation from the entorhinal cortex to anterior and posterior brain regions that overlap with the AT and PM memory systems has been shown to relate to connectivity strength as measured by functional connectivity in cognitively normal adults, consistent with the idea that tau spreads via neuronal connections (*Adams et al., 2019*). Furthermore, tau pathology is related to worse performance in object memory tasks involving the AT system relative to scene memory tasks involving the PM system in cognitive normal older adults, also matching the proposed spread of the disease via neuronal connections, as tau burden was much higher in the AT system compared with the PM system in the early stages of AD (*Arnsten et al., 2020*; *Maass et al., 2019*). While single measures of functional and structural connectivity can give clues into the spread of Ptau, abnormalities in the longitudinal integrity of the structural connectivity may directly reflect loss of axonal integrity from tau pathology (*Conturo et al., 1999*). Especially during predementia AD stages when AD pathology is relatively mild, the loss of this integrity due to tau pathology might help explain how tau accumulation in the MTL leads to memory deficits. Synthesizing these separate lines of evidence, we suspect that the longitudinal integrity of the structural connections of the MTL may link Ptau accumulation and episodic memory decline in predementia AD syndromes.

Here, we capture the longitudinal integrity of the structural connectivity by calculating correlations across multiple time points to give a measure of stability. This novel concept has been first applied to functional connectivity (*Kaufmann et al., 2018*; *Kaufmann et al., 2017*). An emerging study suggests that a functional connectome stability measure is associated with cognition (average and rate of episodic memory decline) (*Ousdal et al., 2020*). Traditional univariate tract-wise analysis first derives a change score of each tract and then compares it across participants. Conversely, the stability measure calculates the pattern similarity between two time points across multiple connections within participants, in which subtle changes of individual connections may affect the stability score. The stability measure may therefore be suitable to capture subtle structural changes in the predementia AD stage.

We aimed to examine the relationship between structural stability, Ptau, and episodic memory. We extracted data from the Alzheimer's Disease Neuroimaging Initiative (ADNI) database, focusing on participants with cerebrospinal fluid (CSF) Ptau collected at baseline, as well as diffusion tensor imaging (DTI) measure twice, 2 years apart with the baseline measure corresponding to Ptau measurement. In addition, we extracted episodic memory data at baseline (anchored by Ptau measurement) and throughout four additional years.

We had several hypotheses: first, based on the idea that the accumulation of Ptau in MTL in the predementia AD stages causes a loss of integrity in the axons that connect MTL to AT and PM systems, we hypothesized that higher CSF Ptau levels will be associated with decreased MTL-related structural stability. Specifically, we examined stability within the MTL, as well as in MTL-AT and MTL-PM connections. Second, given that tau tangles deposit early in the AT lobe (*Arnsten et al., 2020*; *Braak and Del Tredici, 2015*; *Maass et al., 2019*), we hypothesized that Ptau first affects the stability in the MTL and MTL-AT in the predementia stages, followed by MTL-PM as the disease progresses and tau pathology worsens and spreads to posterior brain regions. Finally, given that MTL and related structures play a central role in episodic memory, we hypothesized that MTL-related structural stability might be associated with tau-related episodic memory decline.

## Results

We extracted data from ADNI participants with CSF Ptau collected at baseline, DTI measure twice, 2 year apart, and longitudinal episodic memory data over 5 years. We defined three age- and education-matched groups based on baseline Ptau and cognitive impairment: Ptau negative cognitively unimpaired (CN Ptau–, n = 26), Ptau positive cognitively unimpaired (CN Ptau+, n = 18), and Ptau positive individuals with mild cognitive impairment (MCI Ptau+, n = 30). Group comparison of baseline characteristics is presented in *Table 1*. There was no significant difference between the CN Ptau– and CN Ptau+ groups, except for the Ptau pathology (p < 0.001). Compared with CN groups, MCI Ptau+ group had more APOEε4 carriers, higher Ptau levels, lower CSF Aβ level, greater neurodegeneration, as well as worse performance in episodic memory and global cognition (all ps < 0.05).

Longitudinal integrity of structural connections was measured as a stability index of correlations across baseline and 2 year follow-up for connections between the MTL and cortical memory systems (see Materials and methods for details). We identified regions of interests based on the Desikan-Killiany Atlas. Since the hippocampus and the surrounding hippocampal region including the parahippocampal cortex and entorhinal cortex are the primary regions supporting memory processing (*Davachi et al., 2003*), we consider them to belong to MTL memory network. Thus, the MTL system includes entorhinal, hippocampus, and parahippocampal gyrus in both hemispheres. In line with previous literature (*Berron et al., 2020*; *Ranganath and Ritchey, 2012*), the AT system includes

**Table 1.** Baseline characteristics.

| | CN Ptau–<br>(n = 26) | CN Ptau+<br>(n = 18) | MCI Ptau+<br>(n = 30) | T, F, or $\chi^2$ tests, df1, df2, (p) |
|---|---|---|---|---|
| Age baseline, mean (SD) | 74.0 (4.63) | 74.8 (6.04) | 73.9 (7.34) | 0.14, 2, 71 (0.87) |
| Age baseline ≥ 75, N (%) | 12 (46.2) | 8 (44.4) | 15 (50.0) | 0.16, 2 (0.92) |
| Male, N (%) | 11 (42.3) | 7 (38.9) | 18 (60.0) | 2.65, 2 (0.27) |
| Education, mean (SD) | 16.69 (2.57) | 17.28 (2.42) | 15.67 (2.35) | 2.68, 2, 71 (0.08) |
| APOEε4 carrier, N (%) | 5 (19.2)[a] | 7 (38.8)[a] | 22 (73.3)[b] | **14.79, 2 (<0.001)** |
| CSF Ptau baseline, mean (SD) | 15.8 (4.30)[a] | 29.0 (7.63)[b] | 38.0 (12.34)[c] | **41.58, 2, 71 (<0.001)** |
| CSF Aβ baseline, mean (SD) | 1231.1 (571.89)[a] | 1396.2 (690.95)[a] | 890.7 (383.11)[b] | **5.63, 2, 71 (0.005)** |
| CSF Aβ-positive baseline, N (%) | 9 (34.6)[a] | 7 (38.9)[a] | 22 (73.3)[b] | **9.84, 2 (0.007)** |
| Neurodegeneration baseline, mean (SD) | 2.97 (0.11)[a] | 2.89 (0.17)[a] | 2.77 (0.20)[b] | **9.68, 2, 71 (<0.001)** |
| Neurodegeneration-positive baseline, N (%) | 2 (7.69)[a] | 3 (16.67)[a] | 14 (46.67)[b] | **12.10, 2 (0.002)** |
| Episodic memory baseline, mean (SD) | 0.75 (0.81)[a] | 0.67 (0.79)[a] | 0.14 (0.84)[b] | **4.60, 2, 71 (0.013)** |
| MOCA baseline, mean (SD) | 25.92 (2.07)[a] | 25.89 (2.54)[a] | 21.67 (2.17)[b] | **32.17, 2, 71 (<0.001)** |

Note: CN, cognitively normal; MCI, amnestic mild cognitive impairment; APOEε4, apolipoprotein E ε4; SD, standard deviation; CSF, cerebrospinal fluid; Aβ, beta-amyloid-(1–42); Ptau, phosphorylated tau; MOCA, Montreal Cognitive Assessment. a, b, c represents the post hoc comparison difference from the F-test or $\chi^2$ test. Bold values indicate p < 0.05.

The online version of this article includes the following source data for Table 1:

**Source data 1.** Baseline characteristics.

bilateral inferior temporal, temporal polar, and lateral and medial orbitofrontal cortex, while the PM system includes bilateral posterior and isthmus cingulate, lateral occipital cortex, precuneus, and thalamus. *Figure 1A* shows the center of mass of regions of interests. For visualization purposes, we presented the connections generated between MTL-related structures in a representative participant in *Figure 1B*. The connections within MTL include the hippocampal cingulum bundle and fornix. The connections between MTL and AT largely overlap with the anterior segments of the inferior longitudinal fasciculus (ILF) and inferior fronto-occipital fasciculus (IFOF), while the connections between MTL and PM mainly involve the cingulum bundle and posterior segments of ILF and IFOF.

## Relationship between baseline CSF Ptau and MTL structural stability

We hypothesized that Ptau affects the stability in MTL and MTL-AT early in predementia AD stages (e.g., CN Ptau+) and with disease progression and increased tau pathology (e.g., MCI Ptau+), the stability in MTL-PM is also disrupted. To test this, one-tailed independent t-tests were conducted between (1) CN Ptau+ and CN Ptau–; (2) MCI Ptau+ and CN Ptau–; and (3) MCI Ptau+ and CN Ptau +. We expected to see disrupted stability in MTL and MTL-AT in CN Ptau+, compared with CN Ptau–. For MCI Ptau+, the disruption would extend to MTL-PM, compared to both CN Ptau– and CN Ptau+. One-tailed tests were used because our hypotheses were directional, expecting worse stability with disease progression.

In line with our hypothesis that MTL and MTL-AT stability is affected by Ptau early in the predementia AD stage, compared to CN Ptau–, structural stability was significantly lower in CN Ptau+ in MTL ($t(42) = -2.11$, $p = 0.022$) and MTL-AT ($t(42) = -2.57$, $p = 0.007$). Decreased structural stability was further found in MTL ($t(54) = -2.61$, $p = 0.006$) and MTL-AT ($t(54) = -3.55$, $p < 0.001$) in MCI Ptau+, compared to CN Ptau-. In line with our hypothesis that MTL-PM is affected later in disease progression, we found a significantly lower stability in MTL-PM in MCI Ptau+, compared to CN Ptau + ($t(46) = -5.12$, $p < 0.001$) and CN Ptau– ($t(54) = -3.89$, $p < 0.001$, *Figure 2A*). These results suggested that Ptau affects the structural integrity of connections from the MTL in a way that follows the proposed spread of the disease via neuronal connections.

In addition, using the entire sample, Model 1 (Y $stability = \beta 10 + \beta 11 Ptau + \varepsilon_1$) suggested that levels of Ptau significantly related to structural stability in MTL (B = –0.61, SE = 0.23, Wald's $\chi^2$ = 7.32, BH-adjusted p = 0.011, *Figure 2B*, left), MTL-AT (B = –0.59, SE = 0.21, Wald's $\chi^2$ = 7.73, BH-adjusted p = 0.011, *Figure 2B*, middle), and MTL-PM (B = –0.37, SE = 0.17, Wald's $\chi^2$ = 4.43, BH-adjusted p = 0.035, *Figure 2B*, right).

## The effect of age, CSF Aβ pathology, and neurodegeneration on MTL structural stability in the whole sample

We examined whether structural stability could be directly affected by covariates (i.e., age, CSF Aβ, or neurodegeneration) in the entire sample with Model 2 (Y $stability = \beta 20 + \beta 21 Covariates + \varepsilon_2$). We found no significant relationship between age and stability (all ps > 0.05, *Figure 3A*). We found a significant correlation between CSF Aβ and stability in MTL-PM (β = 0.40, SE = 0.16, Wald's $\chi^2$ = 6.04, BH-adjusted p = 0.042, *Figure 3B*). Neurodegeneration significantly related to stability within MTL (β = 0.58, SE = 0.26, Wald's $\chi^2$ = 4.97, BH-adjusted p = 0.039), MTL-AT (β = 0.49, SE = 0.25, Wald's $\chi^2$ = 3.89, BH-adjusted p = 0.048), and MTL-PM (β = 0.61, SE = 0.19, Wald's $\chi^2$ = 10.39, BH-adjusted p = 0.003, *Figure 3C*).

## The effect of age, CSF Aβ pathology, and neurodegeneration on the relationship between Ptau and the MTL structural stability in the whole sample

We further examined whether the relationship between Ptau and stability would be affected by covariates (i.e., age, Aβ, or neurodegeneration). We tested Ptau and each covariate's interaction effect, controlling for their main effects, in predicting the structural stability in the whole sample with Model 3 (Y $stability = \beta 30 + \beta 31 Ptau + \beta 32 Covariate + \beta 33 Ptau \times Covariates + \varepsilon_3$). Results showed no significant interaction for Ptau × age, Ptau × CSF Aβ, or Ptau × neurodegeneration (*Figure 4*), suggesting that the relationship between Ptau and stability is not affected by age, Aβ pathology, or neurodegeneration.

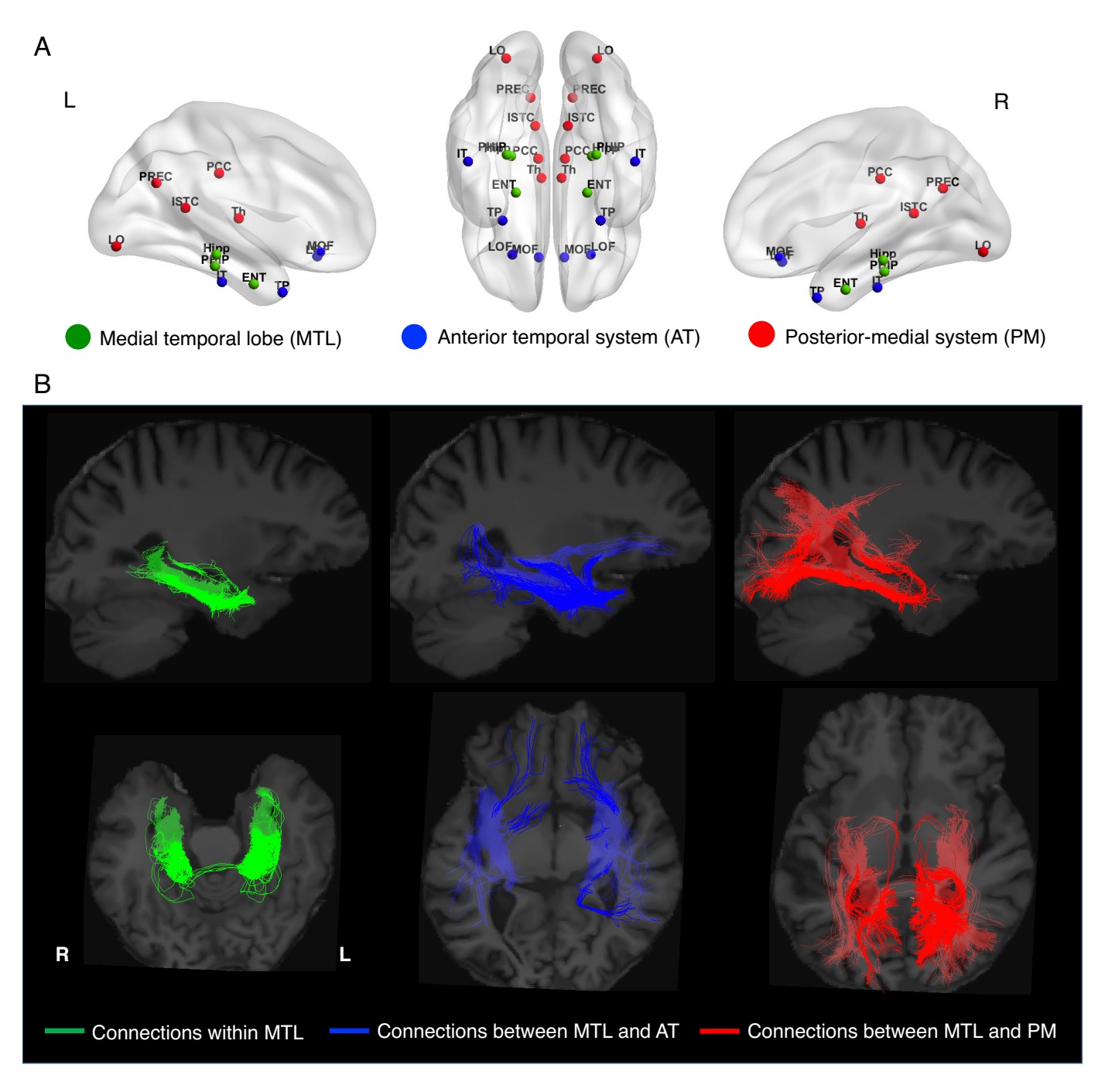

**Figure 1.** Regions of interests. (**A**) Regions of interests are identified based on the Desikan-Killiany Atlas and visualized with the BrainNet Viewer (***Xia et al., 2013***). The medial temporal lobe (MTL) includes bilateral entorhinal cortex (ENT), hippocampus (HIP), and parahippocampal gyrus (PHIP). The anterior-temporal (AT) system includes bilateral inferior temporal cortex (IT), temporal pole (TP), and lateral and medial orbitofrontal cortex (LOF/MOF), while the posterior-medial (PM) system includes bilateral posterior and isthmus cingulate (PCC/ISTC), lateral occipital cortex (LO), precuneus (PREC), and thalamus (TH). (**B**) Visualization of connections between MTL-related structures in a representative participant with TractVis (***Wang and Wedeen, 2007***).

The online version of this article includes the following source data and figure supplement(s) for figure 1:

**Figure supplement 1.** Average mean diffusivity (MD) matrices for each group at each time point.

**Figure supplement 1—source data 1.** Average mean diffusivity (MD) matrices for each group at each time point.

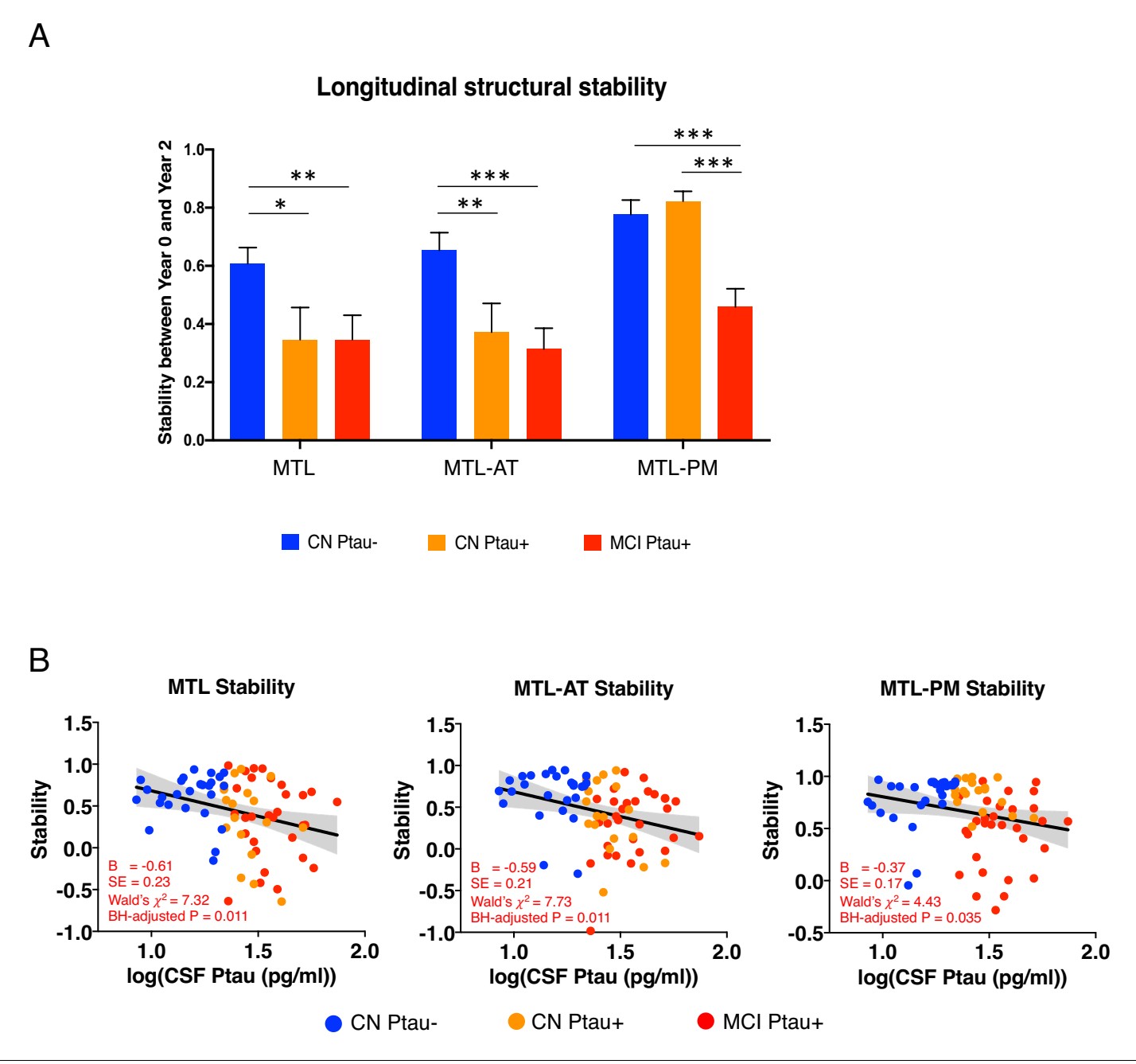

**Figure 2.** Relationship between baseline CSF Ptau and MTL structural stability. (**A**) Structural stability within MTL, MTL-AT, and MTL-PM for CN Ptau–(blue), CN Ptau+ (orange), and MCI Ptau+ (red) groups. Asterisk represents significant group comparison. *p < 0.05; **p < 0.01; ***p < 0.001. (**B**) Relationship of baseline CSF Ptau levels and structural stability within MTL, MTL-AT, and MTL-PM for the entire sample.

The online version of this article includes the following source data and figure supplement(s) for figure 2:

**Source data 1.** Group comparisons.
**Figure supplement 1.** Group comparisons without the exclusion of eight participants.

## Relationship between MTL structural stability, Ptau, and memory over 5 years in the whole sample

Having determined that Ptau related to our measure of structural stability in a way that suggests it follows proposed models of disease progression, we next assessed whether structural stability predicts episodic memory. In line with our hypothesis that decreased stability in MTL-related structures

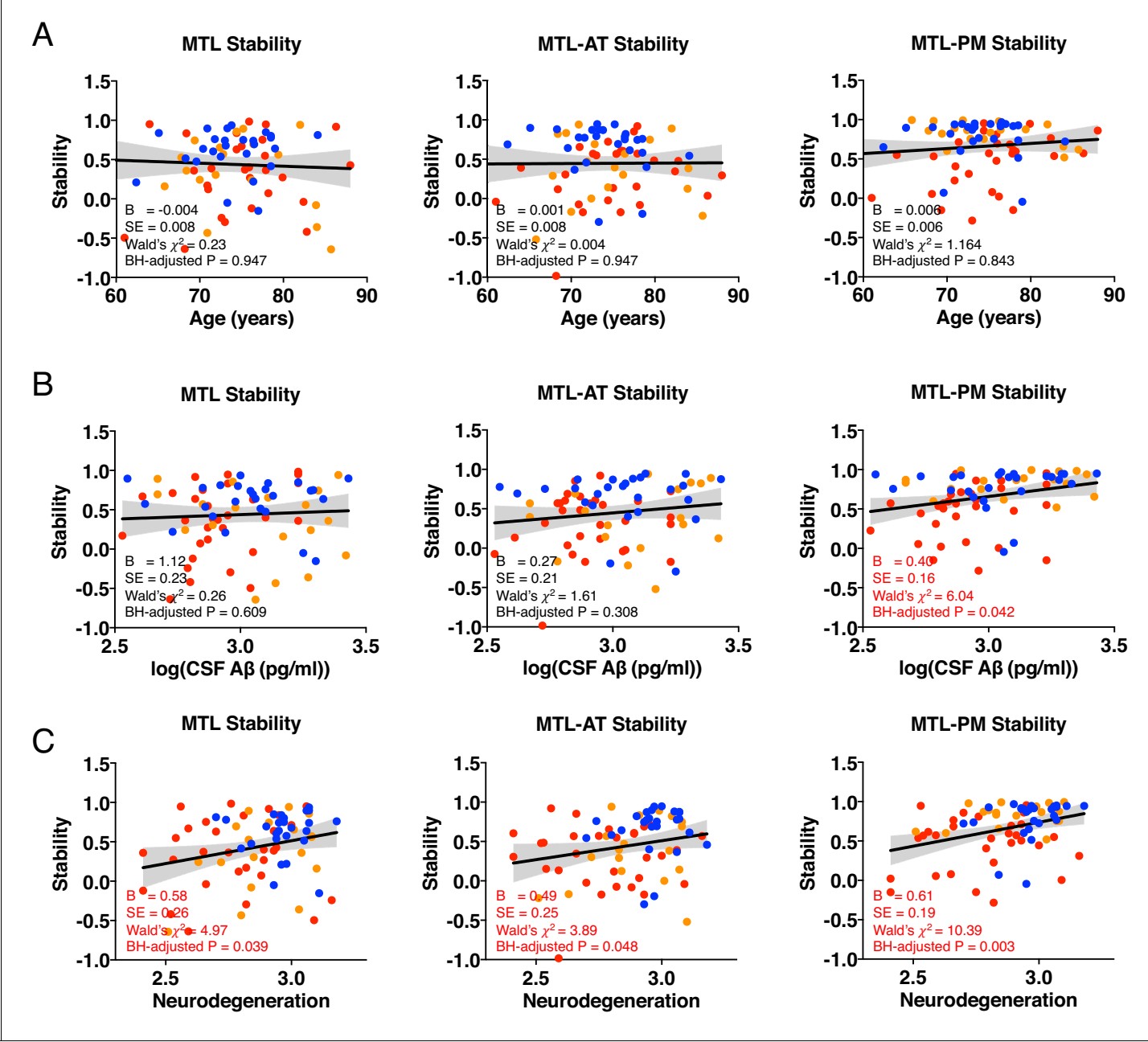

**Figure 3.** The effect of age (A), CSF Aβ pathology (B), and neurodegeneration (C) on the structural stability within MTL, MTL-AT, and MTL-PM in the whole sample. N: neurodegeneration; higher values indicate greater cortical thickness and lower severity in neurodegeneration.

The online version of this article includes the following source data for figure 3:

**Source data 1.** The effect of covariates on the structural stability in the whole sample.

leads to deficits in episodic memory, results from Model 4 (Y $average\ episodic\ memory$ = β40 + β41 $Stability$ + ε$_4$) suggested that stability in MTL (B = 0.60, SE = 0.24, Wald's $\chi^2$ = 6.23, BH-adjusted p = 0.020, *Figure 5A*, left), MTL-AT (B = 0.56, SE = 0.26, Wald's $\chi^2$ = 4.73, BH-adjusted p = 0.030, *Figure 5A*, middle), and MTL-PM (B = 0.95, SE = 0.31, Wald's $\chi^2$ = 9.29, BH-adjusted p = 0.006, *Figure 5A*, right) significantly predicted average episodic memory across 5 years. When we included the covariates (i.e., age, education, sex, neurodegeneration, and CSF Aβ), all results remained significant (MTL: B = 0.42, SE = 0.21, Wald's $\chi^2$ = 3.98, BH-adjusted p = 0.046; MTL-AT: B = 0.47, SE = 0.22, Wald's $\chi^2$ = 4.52, BH-adjusted p = 0.046; MTL-PM: B = 0.88, SE = 0.29, Wald's $\chi^2$ = 9.10, BH-

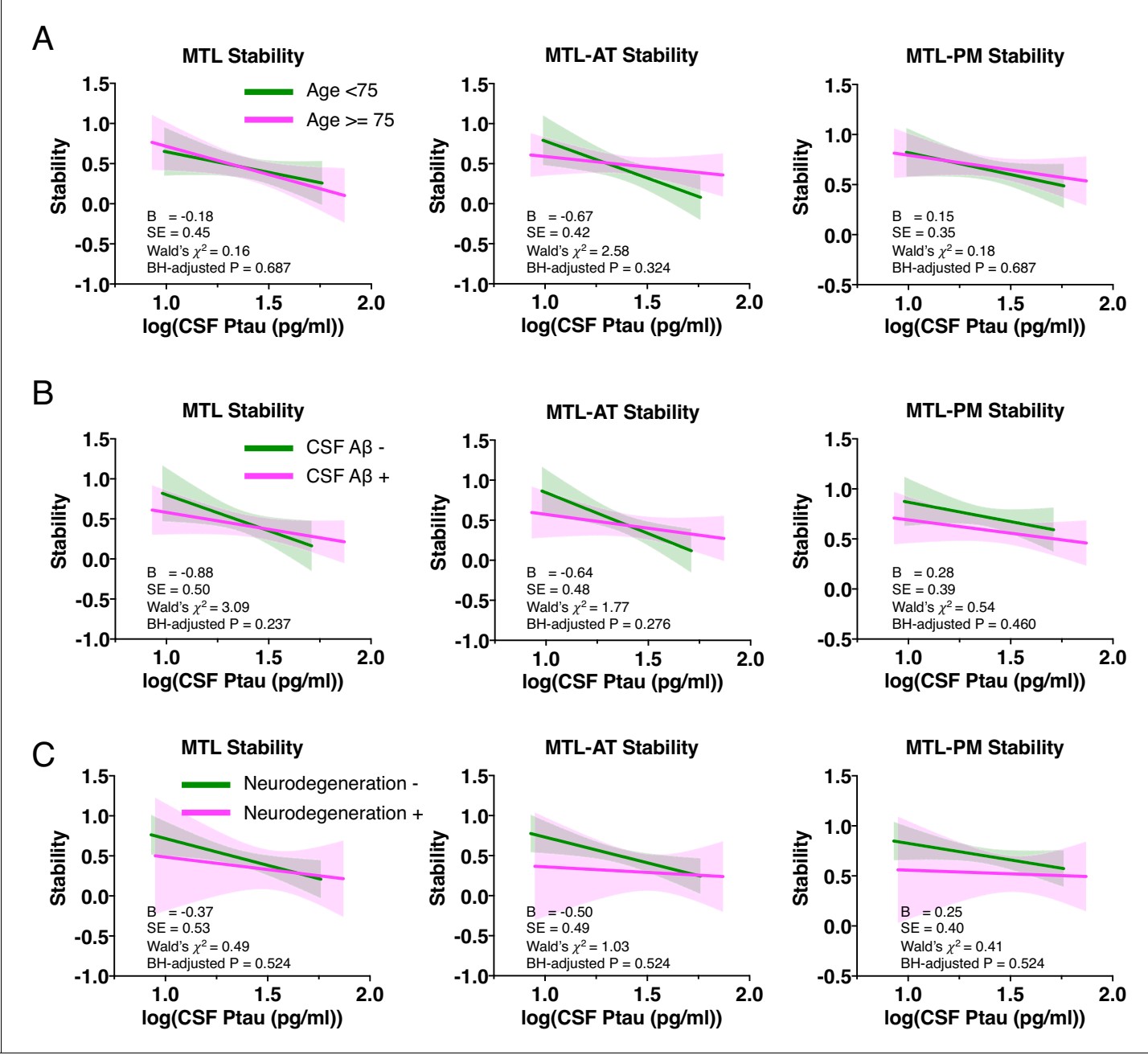

**Figure 4.** The effect of age (**A**), CSF Aβ pathology (**B**), and neurodegeneration (**C**) on the relationship between baseline CSF Ptau levels and the structural stability within MTL, MTL-AT, and MTL-PM in the whole sample.
The online version of this article includes the following source data for figure 4:

**Source data 1.** The effect of covariates on the relationship between Ptau and structural stability.

adjusted p = 0.009). Furthermore, results from Model 4' (Y *rate of episodic memory change* = β40' + β41' *Stability* + ε_4') showed lower stability in MTL (B = 0.12, SE = 0.06, Wald's $\chi^2$ = 4.13, BH-adjusted p = 0.040, *Figure 5B*, left) and MTL-PM (B = 0.15, SE = 0.07, Wald's $\chi^2$ = 4.04, BH-adjusted p = 0.042, *Figure 5B*, right) related to greater decreased rate of episodic memory. We did not find any relationship between MTL-AT stability and rate of episodic memory change (*Figure 5B*, middle). Only the result in MTL remained significant after the covariates were controlled (B = 0.11, SE = 0.06, Wald's $\chi^2$ = 3.85, p = 0.050).

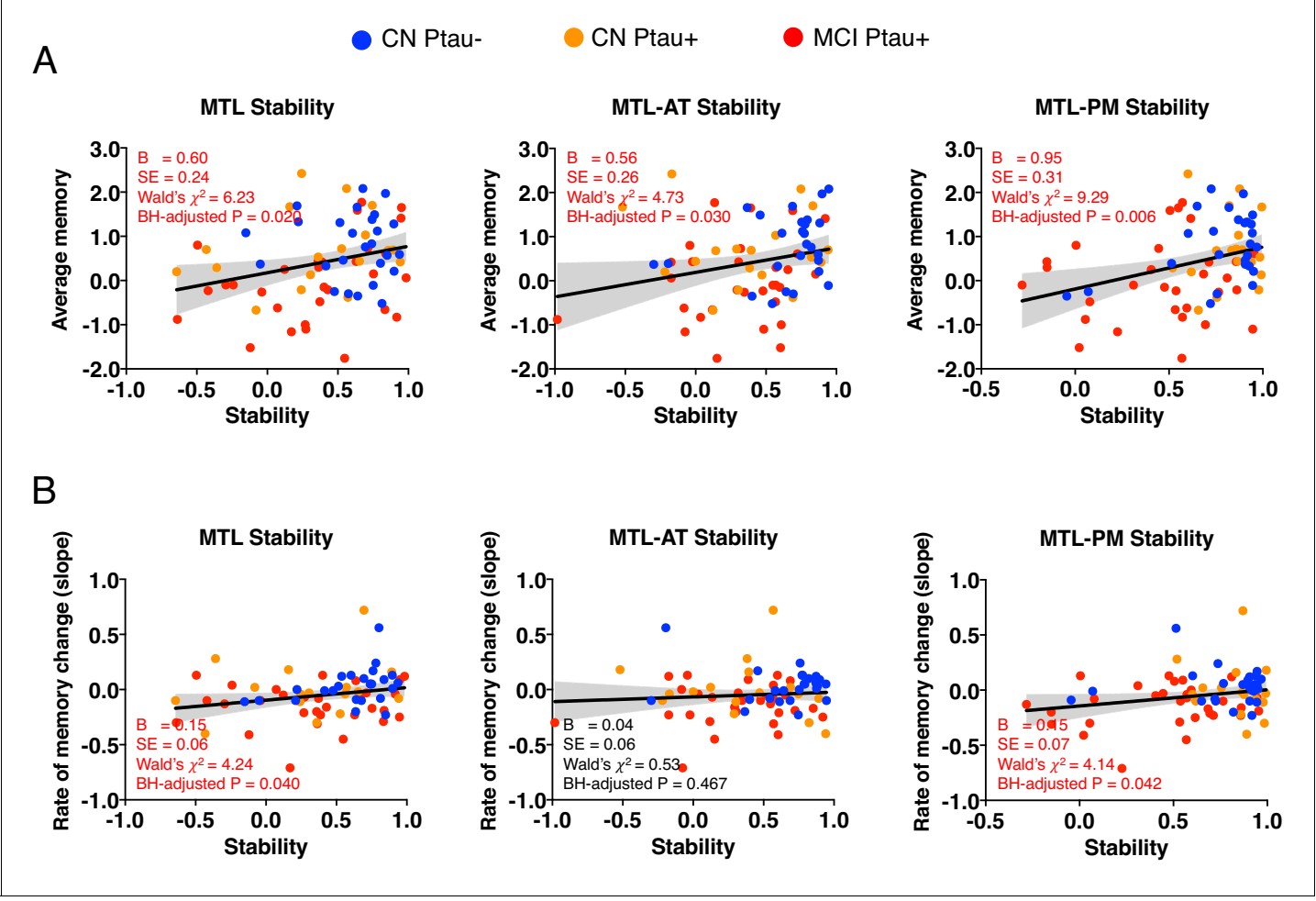

**Figure 5.** MTL structural stability predicts episodic memory. (A) Relationship of average episodic memory over 5 years and the structural stability in the whole sample. (B) Relationship of episodic memory decline rate and the structural stability in the whole sample.

The online version of this article includes the following source data for figure 5:

**Source data 1.** MTL structural stability predicts episodic memory.

To this point, our analyses have established that levels of Ptau significantly related to structural integrity in a manner that follows proposed disease progression, with Ptau in patients early in the disease relating to a loss of structural stability in MTL and MTL-AT connections, followed by MTL-PM connections over time as the disease progresses to MCI. We have also shown that decreased stability in the MTL, MTL-AT, and MTL-PM connections related to worse average episodic memory. Specifically, decreased stability within the MTL connections predicted future rate of memory decline. Our final analysis sought to formally test the hypothesis that this loss of structural stability in MTL provides a link between Ptau accumulation and memory decline. We expected to see stability as a significant moderator of the relationship between Ptau and episodic memory.

Results from Model 5 (Y *average episodic memory* = $\beta 50 + \beta 51$ *Ptau* $+ \varepsilon_5$) suggested that Ptau at baseline predicted average episodic memory across 5 years (B = –1.09, SE = 0.49, Wald's $\chi^2$ = 4.85, p = 0.028). After adding stability and the interaction between stability and Ptau (Model 6: Y *average episodic memory* = $\beta 60 + \beta 61 Ptau + \beta 62 Stability + \beta 63 Ptau \times Stability + \varepsilon_6$), there was no significant interaction between Ptau and stability of MTL (B = 1.34, SE = 1.62, Wald's $\chi^2$ = 0.69, p = 0.41).

Results from Model 5' (Y rate of *episodic memory change* = $\beta 50' + \beta 51'$ *Ptau* $+ \varepsilon_{5'}$) suggested that higher level of Ptau predicted greater decreased rate of memory (B = –0.38, SE = 0.11, Wald's $\chi^2$ = 11.34, p = 0.001). After adding stability and the interaction between stability and Ptau into the model (Model 6': Y rate of *episodic memory change* = $\beta 60' + \beta 61' Ptau + \beta 62' Stability + \beta 63' Ptau \times$

*Stability* + $\varepsilon_{6'}$), Ptau × Stability of MTL was significant (B = –0.01, SE = 0.01, Wald's $\chi^2$ = 5.28, p = 0.022). The interaction effect remained significant while accounting for covariates (B = –0.01, SE = 0.01, Wald's $\chi^2$ = 3.97, p = 0.046). *Figure 6* displayed the associations between Ptau and rate of memory change depending on MTL stability. Participants were divided based on terciles of MTL stability. Stronger relationship between Ptau and rate of memory change was observed in those with

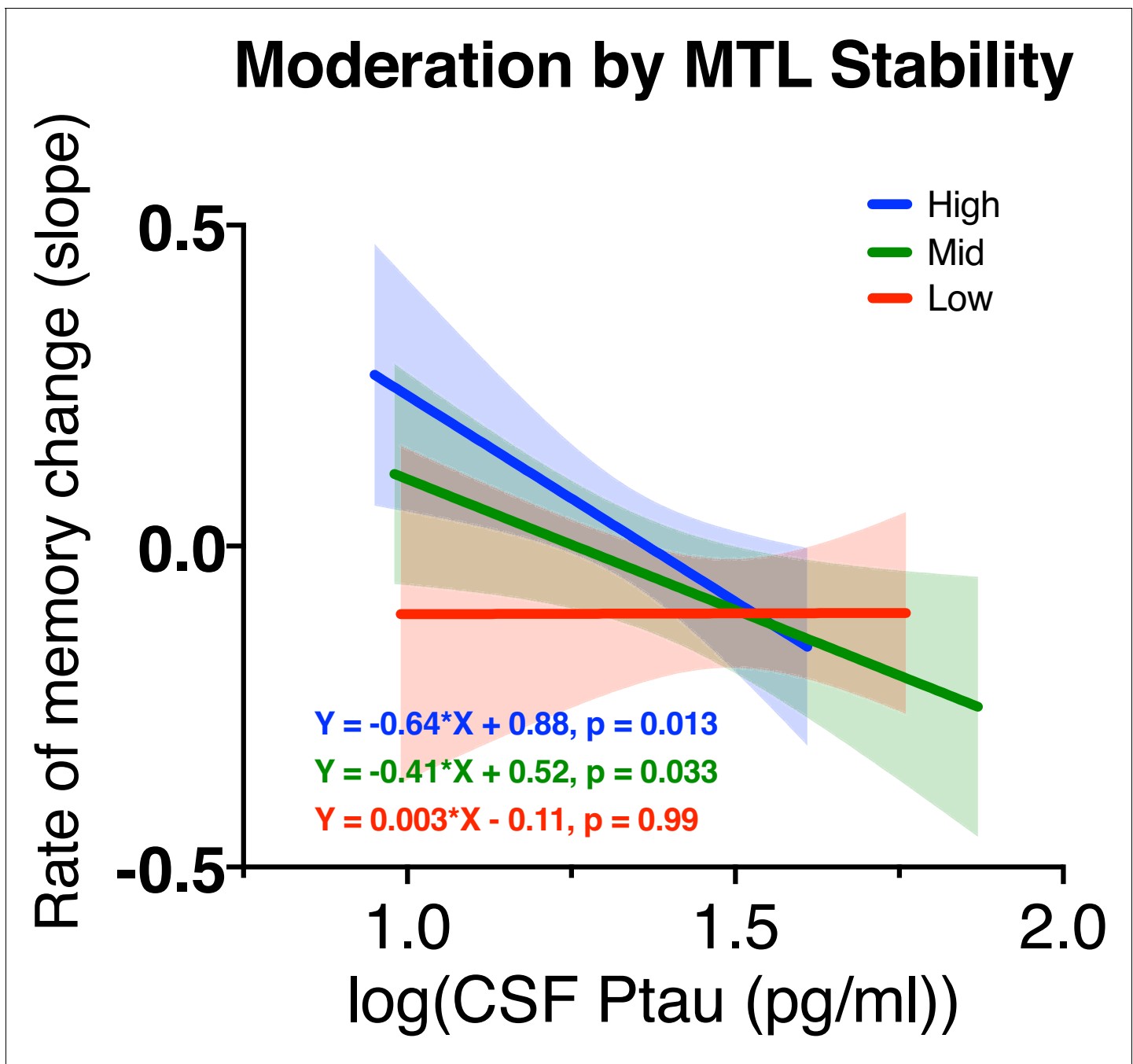

**Figure 6.** The moderating effect of medial temporal lobe (MTL) structural stability on the relationship between Ptau and rate of memory change. The plot shows how structural stability within MTL affected the relationship between Ptau and rate of memory change. Participants were divided into terciles based on their MTL stability. The lowest tercile is shown in red, the middle in green, and the highest in blue. P-values indicate whether the slope of the regression line is significantly different from zero for each tercile separately.

The online version of this article includes the following source data for figure 6:

**Source data 1.** MTL structural stability moderates the relationship between Ptau and memory change.

higher MTL stability. All of these results suggested that the structural stability of the MTL links Ptau and episodic memory.

## Discussion

In this study, we examined whether the presence of CSF Ptau at baseline was related to a loss of structural stability in a way that matched proposed disease progression and whether this loss of stability in connections important for episodic memory is related to the link between Ptau accumulation and cognitive decline. We did this by first examining the relationship between Ptau at baseline and longitudinal white matter stability across three groups (CN Ptau–, CN Ptau+, and MCI Ptau+). CN Ptau+ and MCI Ptau+ had decreased stability in MTL and MTL-AT compared to CN Ptau-, while MCI Ptau+ further showed decrease in MTL-PM stability relative to both CN Ptau– and CN Ptau+ groups, suggesting that Ptau relates to structural stability in a temporal pattern that matches its proposed spread from the MTL via neuronal connections. Furthermore, the stability in MTL, MTL-AT, and MTL-PM predicted average episodic memory over 5 years, suggesting that damage to these pathways specifically relates to episodic memory decline. Importantly, structural stability in MTL moderated the effect of Ptau on the rate of memory change, suggesting that structural stability in MTL may be a candidate as a mechanistic link between the accumulation of Ptau and cognitive decline.

### Ptau influences the white matter structural stability

Comparing groups with varied degrees of Ptau abnormality (Ptau+ vs. Ptau–) and/or cognitive deficits (CN vs. MCI), MTL structural stability was disrupted both within MTL and between MTL and AT in CN Ptau+ and MCI Ptau+ relative to CN Ptau–, and the disruption was further extended to MTL-PM connections in MCI Ptau+ relative to both CN Ptau– and CN Ptau+. This is consistent with the proposed model of Ptau spread from the MTL via neuronal connections. In a recent longitudinal study examining Ptau's influence on the structural connectome in the preclinical stage of AD, elevated Ptau was associated with changes primarily in MTL and MTL-AT regions (*Kim et al., 2019*). Another study found that Ptau accumulation in posterior cingulate cortex (a region of PM) could be predicted by baseline mean diffusivity (MD) in MTL (*Jacobs et al., 2018*). Noticeably in our study, the relationship between Ptau and structural stability was robust, as it was barely affected by neurodegeneration, Aβ, or age. In summary, our results add to previous literature suggesting that tau pathology may act as the primary mechanism for changes in white matter structural integrity.

### White matter structural stability predicts longitudinal episodic memory

In line with previous findings (*Jacobs et al., 2018*), our study revealed a robust relationship between structural stability in MTL, MTL-AT, and MTL-PM connections and average episodic memory. Additionally, stability of MTL related to the decline rate of memory over time. Literature suggests that the decline in episodic memory can be predicted by changes in functional connectivity between select regions (e.g., cortico-striatal [*Fjell et al., 2016*], subcortical [*Ousdal et al., 2020*], default mode networks [*Staffaroni et al., 2018*], and posterior medial temporal lobe [*Salami et al., 2016*]). Our findings add to this literature by showing that an index reflecting the longitudinal integrity of MTL connections relates to episodic memory decline over time, suggesting that changes in structural integrity of within MTL connections over the course of several years co-occur with episodic memory decline. These results suggested that MTL-related structural connections might serve as potential therapeutic targets for preventing memory decline in the early stage of AD.

### MTL structural stability links Ptau and longitudinal episodic memory

Previous research has shown that MTL structural connectivity influences downstream PM Ptau, and changes in both MTL white matter and PM Ptau influenced memory (*Jacobs et al., 2018*). However, it is unclear whether changes in MTL structural connectivity are a result of accumulated Ptau and whether these changes present a mechanism by which Ptau may lead to decreases in episodic memory. In the current study, we found that levels of Ptau were related to decreases in MTL structural stability and that these changes were in turn related to decreases in episodic memory. A model showed that decreases in the stability of within MTL connections significantly predicted episodic memory decline, suggesting that changes in these connections resulting from increased Ptau are associated with decreases in episodic memory. Synthesizing the relationship between Ptau and

structural stability, as well as structural stability and episodic memory, we found MTL structural stability moderated the relationship between Ptau and rate of memory change. Ptau's predictive strength for cognition is overall limited, which interferes with its clinical value for early detection of cognitive decline (*Hansson and Mormino, 2018*). Structural stability in MTL may be a viable biomarker and early intervention target addressing the linkage between pathology and cognition. It is possible that the observation of a stronger relationship between more Ptau and faster rate of memory decline in those with higher MTL stability might be driven by greater variance in this group, caused by the inclusion of both stable individuals with overall low connectivity as well as consistently high connectivity. To assess this, we looked at the number of spread of individuals across these groups based on their mean structural connectivity in MTL (*Table 2*, see Materials and methods for details). The stable group contains 12 high–high MD and 15 low–low MD subjects, while the instable group contains 12 high–high MD and nine low–low MD subjects. While there are more low-low subjects in the stable group, the ratio of low-low to high–high is similar across these groups, suggesting increased variance is unlikely to drive the strong relationship with cognition in the high-stability group.

### Limitations

This study has several limitations. The major limitation of this study was related to its small sample size, since we have only 18 participants in the CN Ptau+ group. In future studies with larger sample size, data-driven analyses could be conducted to validate our findings and explore other relevant findings. Second, Ptau is determined using one-time measurement from CSF rather than longitudinal PET. All regionally specific hypotheses on tau accumulation are based on previous studies, rather than observations in the current study. Finally, the 2 year interval for DTI data was relatively short, which may not be sufficient to capture reliable changes in structural connectivity in predementia AD. Longer follow-up period may reveal the significant interaction effect between Ptau and covariates (e. g., Aβ) in predicting the structural stability.

## Materials and methods

### Data source

Data were obtained from ADNI GO and ADNI2 database (adni.loni.usc.edu). The ADNI was launched in 2003 as a public–private partnership, led by Principal Investigator Michael W. Weiner, MD. The primary goal of ADNI has been to test whether serial magnetic resonance imaging (MRI), PET, other biological markers, and clinical and neuropsychological assessment can be combined to measure the progression of MCI and early AD. For up-to-date information, see http://www.adni-info.org. ADNI GO and ADNI2 collect diffusion-weighted images (DWI) at 14 sites across North

**Table 2.** Dichotomized mean MD and stability of MTL.

| Mean MD of MTL at year 0 | Mean MD of MTL at year 2 | Stability of MTL | CN Ptau– (N = 26) | CN Ptau+ (N = 18) | MCI Ptau+ (N = 30) |
|---|---|---|---|---|---|
| High | High | Stable | 8 (30.8%) | 0 (0%) | 4 (13.3%) |
| High | Low | Stable | 0 (0%) | 0 (0%) | 3 (10.0%) |
| Low | High | Stable | 6 (23.1%) | 1 (5.6%) | 0 (0%) |
| Low | Low | Stable | 4 (15.4%) | 7 (38.9%) | 4 (13.3%) |
| High | High | Instable | 2 (7.6%) | 6 (33.3%) | 4 (13.3%) |
| High | Low | Instable | 0 (0%) | 2 (11.1%) | 8 (26.7%) |
| Low | High | Instable | 3 (11.5%) | 0 (0%) | 3 (10.0%) |
| Low | Low | Instable | 3 (11.5%) | 2 (11.1%) | 4 (13.3%) |

Note: Mean MD is the averaged MD across all connections within MTL. MD, mean diffusivity; MTL, medial temporal lobe.

The online version of this article includes the following source data for Table 2:

**Source data 1.** Dichotomized mean MD and stability of MTL.

America, all using the same scanner manufacturer (General Electric), magnetic field strength (3T) protocol, including the same voxel size and gradient directions. To reduce site-related errors, ADNI group made rigor standardization and quality control over the protocols for patient recruitment and imaging data acquisition across sites. Each exam underwent a quality control evaluation at the Mayo Clinic (Rochester, MN), including inspection of each incoming image file for protocol compliance, clinically significant medical abnormalities, and image quality (*Jack et al., 2018*). ADNI3 was not used here due to different scan protocols and limited number of eligible participants at the time of manuscript preparation.

## Participants

All participants in this study were from ADNI GO and ADNI2 where DTI data were collected using 3T GE scanners. Only non-demented individuals (healthy controls and MCI) with CSF pTau data at baseline and DTI data at two time points (2 years apart) were selected for this study. Participants were classified into different groups according to their CSF Ptau status and their clinical diagnosis, which resulted in the following participants: 29 Ptau negative cognitively unimpaired (CN Ptau–), 20 Ptau positive cognitively unimpaired (CN Ptau+), and 36 Ptau positive with MCI (MCI Ptau+). We then excluded three participants from the imaging data analysis due to inadequate tractography quality in either time of DTI data (none from CN Ptau–, one from CN Ptau+, and two from MCI), and eight participants to optimize age and education matching across groups (three participants aged <65 from CN Ptau–, one at age 89 from CN Ptau+, and four with years of education $\leq$ 12 from MCI Ptau+). The final sample contained 74 participants (26 CN Ptau–, 18 CN Ptau+, and 30 MCI Ptau+). Participant characteristics were provided in *Table 1*. The main result remains the same without the exclusion of eight participants (see *Figure 2—figure supplement 1*).

## Measures

### CSF biomarkers

Ptau data were derived from CSF aliquots and measured using Elecsys CSF immunoassays at the University of Pennsylvania. The data is available in the 'UPENNBIOMK10.csv' file in the ADNI database. The cut-off for positive Ptau pathology was >21.8 pg/ml (*Suárez-Calvet et al., 2019*). Of note, Ptau data did not follow a normal distribution and were hence log10-transformed when used as a continuous variable.

## Imaging data

### Acquisition

All participants underwent whole-brain MRI scanning on 3T GE scanners. DWI were collected with the following parameters: matrix size = 256 × 256 mm; flip angle = 90°; slice thickness = 2.7 mm; 41 diffusion-weighted images (b = 1000 s/mm$^2$) and five non-diffusion-weighted b0 image. T1-weighted spoiled gradient recalled echo sequences were acquired in the same scanning session (TR = 7.0–7.7 ms, TE = 2.8–3.2 ms, TI = 400 ms, matrix size = 256 × 256 mm, flip angle = 11°, resolution 1.02 × 1.02 mm, slice thickness = 1.20 mm). More details on ADNI protocols may be found at http://adni.loni.usc.edu/methods/documents/mri-protocols/.

### Preprocessing

Each raw DWI image was aligned to the average b0 image using the FSL eddy_correct tool 6.0.1 (http://www.fmrib.ox.ac.uk/fsl) to correct for head motion and eddy current distortions. Non-brain tissue was removed using FSL's Brain Extraction Tool (*Smith, 2002*). We then registered DWI images with the T1 anatomical images using Advanced Normalization Tools (ANTS; http://www.picsl.upenn.edu/ANTS/).

### Network construction

To extract brain connectome, we employed an established structural connectome processing pipeline (for details, see *Zhang et al., 2018*). First, we applied a reproducible probabilistic tractography algorithm (*Girard et al., 2014*) to diffusion MRI data to generate streamlines across the whole brain. We used 0.2 mm step size and 25° angle threshold. Next, we defined the cortical, subcortical, and brainstem regions on T1 anatomical images for each participant using FreeSurfer (http://surfer.nmr.

[mgh.harvard.edu/](mgh.harvard.edu/)). From the FreeSurfer segmentation, 34 regions per hemisphere were identified using Desikan-Killiany atlas (*Desikan et al., 2006*), as well as 17 subcortical and brainstem structures: the brain stem, and bilateral segmentations of thalamus, caudate, putamen, pallidum, hippocampus, amygdala, accumbens, and cerebellum. Together, we obtained 85 nodes per participant. For each pair of nodes, we extracted the streamlines connecting them. Next, we calculated the mean of mean diffusivity (MD), an indicator of myelination and axonal thickness, along each streamline to describe the connectivity (edge) between the connected nodes. This procedure generated an 85-by-85 symmetric MD-based structural connectivity matrix per participant at both baseline (year 0) and 2 year follow-up (year 2). Of note, previous studies have shown that MD is more sensitive than other metrics such as fractional anisotropy (FA) to white matter alterations associated with MCI (*Yu et al., 2017*) and AD (*Jin et al., 2017*), with MD also predicting conversion to dementia (*Fellgiebel et al., 2006*; *Müller et al., 2007*). Physiologically, the absolute diffusion, which is quantified by MD, is a more sensitive marker of neurodegeneration than FA, which simply quantifies the anisotropy of the diffusion tensor (*Acosta-Cabronero et al., 2010*). These findings motivated us to choose MD over other metrics.

## Stability measure

For each individual, we computed longitudinal stability by calculating a within-participant Pearson correlation coefficients between year 0 and year 2 across connections. Each individual has stability scores for MTL, MTL-AT, and MTL-PM, respectively: the score for MTL is based on the 15 connections within MTL, MTL-AT based on the 48 unique connections between MTL and AT system, and MTL-PM based on 60 unique connections between MTL and PM system. *Figure 1—figure supplement 1* shows the average mean diffusivity (MD) matrices for each group at year 0 and year 2, respectively.

The traditional univariate analysis only considers the overall magnitude of the network (e.g., the average MD across all connections within MTL network). By contrast, the stability measure is interested in the pattern of connectivity and reflects change in the patterns of MD across connections. This measure is purposefully meant to be agnostic to the mechanism by which stability is impacted across individuals; it is designed to capture changes in the pattern of connectivity that we believe occur in response to pathology. It may be that pathology causes connections to be reduced in strength, but other connections may show corresponding increases as a result. To understand how this measure related to overall connectivity, we averaged the MD measures across all connections within MTL to get a mean MD score for MTL network for each individual. Then we used the median as a cut-off to dichotomize the mean MD score to 'high' and 'low' for baseline (year 0) and 2 year follow-up (year 2), respectively. Similarly, we split stability of MTL to 'stable' and 'instable' at the median. The mean MD scores of 2 years and the MTL stability were dichotomized into two levels, over and under the cut point, forming eight groups in total. *Table 2* displays the prevalence of eight groups in the CN/MCI samples. As we can see, it is possible that subjects maintained the same level of mean MD from baseline to follow-up (e.g., low–low or high–high), but their MD patterns actually changed and became instable. Thus, the traditional univariate analysis may not be as sensitive as the stability measure in detecting these subtle structural changes in the predementia AD stage.

## Episodic memory measure

Episodic memory is a standardized composite score derived from multiple measures using factor analyses (*Crane et al., 2012*), on the memory-related domains of the Mini-Mental Status Examination, Alzheimer's Disease Assessment Scale-Cognition subscale, Rey Auditory Verbal Learning Test, and Logical Memory Test. The average standardized episodic memory composite score is 0 among the entire ADNI sample (including those with cognitive impairment). We had follow-up data for episodic memory for all participants for up to 4 years (from baseline to a 4 year follow-up). To assess the association between longitudinal structural stability, baseline Ptau, and episodic memory, an average episodic memory score is calculated by averaging episodic memory over 5 years. Since we were interested in prediction of memory in the future, average memory score was used instead of baseline memory. Rate of memory change was determined by calculating the slope obtained from the linear regression line created by all available time points.

## Covariates

We consider baseline demographics and factors associated with AD or episodic memory (i.e., age, Aβ, and neurodegeneration) as covariates in selected analysis. The cut-off for age was ≥75 years old since it has been widely used to classify young-old (65–74) and old-old (75+) in previous literature (*Field and Minkler, 1988*). Aβ were derived from CSF aliquots and measured using Elecsys CSF immunoassays at the University of Pennsylvania. The cut-off for positive Aβ was <976.6 pg/ml (*Suárez-Calvet et al., 2019*). A log-data transformation was applied to fit the skewed Aβ distribution into a normal distribution when used as a continuous variable. Neurodegeneration was quantified using structural MRI data by calculation of AD signature cortical thickness, consisting of posterior brain regions, including bilateral inferior and middle temporal lobes, entorhinal cortex, and fusiform gyrus, based on automated cortical parcellation using the Desikan-Killiany Atlas with Freesurfer (*Jack et al., 2015*; *Lin et al., 2017*). Higher values in AD signature cortical thickness indicate greater cortical thickness and lower severity in neurodegeneration. The cut-off for abnormal neurodegeneration was <2.77 mm (*Jack et al., 2015*).

## Statistical analyses

To determine between-group differences in demographic and clinical information, one-way ANOVA and chi-square tests were used for continuous or categorical variables, respectively.

To examine the relationship between baseline CSF Ptau and structural stability, we conducted one-tailed independent t-tests between (1) CN Ptau+ and CN Ptau–; (2) MCI Ptau+ and CN Ptau–; and (3) MCI Ptau+ and CN Ptau+. In addition, we also conducted generalized linear model (GLM) using the entire sample with Ptau as a continuous variable.

Model 1: Y *stability* = $\beta 10 + \beta 11 Ptau + \varepsilon_1$.

We examined whether structural stability could be directly affected by covariates (i.e., age, CSF Aβ or neurodegeneration) using GLMs.

Model 2: Y *stability* = $\beta 20 + \beta 21 Covariates + \varepsilon_2$.

To examine whether the relationship between Ptau and stability would be affected by covariates (i.e., age, Aβ, or neurodegeneration), we tested Ptau and each covariate's interaction effect, controlling for their main effects, in predicting the structural stability using a GLM in the whole sample.

Model 3: Y *stability* = $\beta 30 + \beta 31 Ptau + \beta 32 Covariate + \beta 33 Ptau \times Covariates + \varepsilon_3$.

We examined relationships between longitudinal structural stability, baseline Ptau, and episodic memory in the whole sample, including and excluding the covariates (age, education, sex, neurodegeneration, and CSF Aβ).

First, we expected to see that the stability in MTL-related structures could predict episodic memory. GLMs were conducted with and without covariates.

Model 4 (Stability predicts average episodic memory based on β31): Y *average episodic memory* = $\beta 40 + \beta 41$ *Stability* $(+\beta 42 Covariates) + \varepsilon_4$.

Model 4' (Stability predicts decline rate of episodic memory based on β31'): Y *rate of episodic memory change* = $\beta 40' + \beta 41'$ *Stability* $(+\beta 42' Covariates) + \varepsilon_{4'}$.

Next, we tested whether Ptau could predict episodic memory using GLMs.

Model 5 (Ptau predicts average episodic memory based on β51): Y *average episodic memory* = $\beta 50 + \beta 51$ *Ptau* $+ \varepsilon_5$.

Model 5' (Ptau predicts decline rate of episodic memory over time based on β51'): Y rate of *episodic memory change* = $\beta 50' + \beta 51'$ *Ptau* $+ \varepsilon_{5'}$.

Finally, to examine whether stability moderated the relationship between Ptau and episodic memory, we tested the interaction between Ptau and Stability in model 6 or 6' (β63 and β63'), including and excluding the covariates.

Model 6: Y *average episodic memory* = $\beta 60 + \beta 61 Ptau + \beta 62 Stability + \beta 63 Ptau \times Stability$ $(+\beta 64 Covariates) + \varepsilon_6$.

Model 6': Y rate of *episodic memory change* = $\beta 60' + \beta 61' Ptau + \beta 62' Stability + \beta 63' Ptau \times Stability$ $(+\beta 64' Covariates) + \varepsilon_{6'}$.

The analysis was conducted in SPSS 24.0 (IBM Corp., Armonk, NY). Results were corrected for multiple comparisons across MTL, MTL-AT, and MTL-PM using Benjamini–Hochberg (BH) procedure when appropriate and BH-adjusted p-values were reported.

## Acknowledgements

Data collection and sharing for this project was funded by the Alzheimer's Disease Neuroimaging Initiative (ADNI) (National Institutes of Health Grant U01 AG024904) and DOD ADNI (Department of Defense award number W81XWH-12-2-0012). ADNI is funded by the National Institute on Aging, the National Institute of Biomedical Imaging and Bioengineering, and through generous contributions from the following: AbbVie, Alzheimer's Association; Alzheimer's Drug Discovery Foundation; Araclon Biotech; BioClinica, Inc; Biogen; Bristol-Myers Squibb Company; CereSpir, Inc; Cogstate; Eisai Inc; Elan Pharmaceuticals, Inc; Eli Lilly and Company; EuroImmun; F Hoffmann-La Roche Ltd and its affiliated company Genentech, Inc; Fujirebio; GE Healthcare; IXICO Ltd.; Janssen Alzheimer Immunotherapy Research and Development, LLC.; Johnson and Johnson Pharmaceutical Research and Development LLC.; Lumosity; Lundbeck; Merck and Co., Inc; Meso Scale Diagnostics, LLC.; NeuroRx Research; Neurotrack Technologies; Novartis Pharmaceuticals Corporation; Pfizer Inc; Piramal Imaging; Servier; Takeda Pharmaceutical Company; and Transition Therapeutics. The Canadian Institutes of Health Research is providing funds to support ADNI clinical sites in Canada. Private sector contributions are facilitated by the Foundation for the National Institutes of Health (http://www.fnih.org). The grantee organization is the Northern California Institute for Research and Education, and the study is coordinated by the Alzheimer's Therapeutic Research Institute at the University of Southern California. ADNI data are disseminated by the Laboratory for Neuro Imaging at the University of Southern California.

## Additional information

### Funding

| Funder | Grant reference number | Author |
| --- | --- | --- |
| National Institutes of Health | R01 NR015452 | Feng V Lin |
| National Institutes of Health | U01 AG024904 | Feng V Lin |
| Department of Defense | W81XWH-12-2-0012 | Feng V Lin |

The funders had no role in study design, data collection and interpretation, or the decision to submit the work for publication.

### Author contributions

Quanjing Chen, Conceptualization, Data curation, Formal analysis, Validation, Investigation, Visualization, Methodology, Writing - original draft; Adam Turnbull, Formal analysis, Writing - original draft, Writing - review and editing; Timothy M Baran, Writing - original draft, Writing - review and editing; Feng V Lin, Conceptualization, Supervision, Funding acquisition, Writing - original draft, Writing - review and editing

### Author ORCIDs

Quanjing Chen  https://orcid.org/0000-0003-4630-6817

### Ethics

Human subjects: The current study is a secondary data analysis of limited-identified data per data user agreement between ADNI and F.L. The human subject research of original ADNI data collection was conducted at each ADNI data collection site (see the full list of sites http://adni.loni.usc.edu/wp-content/uploads/how_to_apply/ADNI_Acknowledgement_List.pdf). Written informed consent was obtained from each participant (see http://adni.loni.usc.edu/wp-content/uploads/2008/07/adni2-procedures-manual.pdf for detailed information about ethical procedures for ADNI). Protocol_11.19.14.

### Decision letter and Author response

Decision letter https://doi.org/10.7554/eLife.62114.sa1
Author response https://doi.org/10.7554/eLife.62114.sa2

## Additional files

### Supplementary files
• Transparent reporting form

### Data availability

Data used in preparation of this article were obtained from the Alzheimer's Disease Neuroimaging Initiative (ADNI) database (http://adni.loni.usc.edu/). As such, the investigators within the ADNI contributed to the design and implementation of ADNI and/or provided data but did not participate in analysis or writing of this report. A complete listing of ADNI investigators can be found at: http://adni.loni.usc.edu/wp-content/uploads/how_to_apply/ADNI_Acknowledgement_List.pdf.

The following previously published dataset was used:

| Author(s) | Year | Dataset title | Dataset URL | Database and Identifier |
|---|---|---|---|---|
| Jack CR, Bernstein MA, Fox NC, Thompson P, Alexander G, Harvey D, Borowski B, Britson PJ, Whitwell LJ, Ward C, Dale AM | 2008 | The Alzheimer's disease neuroimaging initiative | https://ida.loni.usc.edu/login.jsp?project=ADNI | ADNI, adni |

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
