## [Decision Letter]

**Acceptance summary:**

Chen and colleagues provide timely new insights into the link between CSF measures of phosphorylated tau, changes in brain structural connectivity over time, and cognitive decline in a sample of older adults who are cognitively normal or have mild cognitive impairment. The concept of "stability" used here provides a novel means by which changes in structural integrity can be assessed, and opens up new avenues to understand how tau impacts structural brain networks implicated in episodic memory decline in aging and Alzheimer's disease.

**Decision letter after peer review:**

Thank you for submitting your article "Longitudinal stability of medial temporal lobe connectivity explains tau-related memory decline" for consideration by *eLife*. Your article has been reviewed by Chris Baker as the Senior Editor, a Reviewing Editor, and three reviewers. The following individual involved in review of your submission has agreed to reveal their identity: Kaitlin Kassady (Reviewer #3).

The reviewers have discussed the reviews with one another and the Reviewing Editor has drafted this decision to help you prepare a revised submission.

Summary:

Chen et al., present a timely study exploring the link between CSF measures of Ptau, changes in brain structural connectivity across 2 years (i.e., "stability"), and cognitive decline in a sample of older adults who are either cognitively normal or have mild cognitive impairment. While the earliest changes in structural connectivity were found within the MTL, and between the MTL and AT system, MTL and MTL-PM stability was related to better average episodic memory and change in episodic memory over five years. The research question of how tau impacts specific structural networks is important to further understand episodic memory impairment in aging and Alzheimer's disease, and the concept of "stability" provides a novel means by which to assess changes in structural integrity.

Essential revisions:

1) Introduction:

a) The authors state that hyperphosphorylated tau proteins are "necessary for subsequent Aβ aggregation". However, the prevailing view in the field is that Aβ aggregation is not dependent on concurrent tau pathology (ex. autosomal dominant AD). Please amend.

b) The "PM" system is referred to as "posterior-MTL", but there are no MTL regions in their PM network. It would be helpful to use the terminology "posterior-medial" when referring to "PM", as well as citing Ranganath and Ritchey, 2012, which first introduced this framework.

d) In the Introduction the authors state that the "MTL is more strongly connected to the AT than PM systems", however this is not accurate. In the papers they cite (especially Navarro Schröder, 2015), some regions of the MTL (such as parahippocampal gyrus and posteromedial entorhinal cortex) are more strongly connected to the PM system.

e) I believe Berron et al., showed mostly reduced MTL-AT functional connectivity in CU Ab+ vs CU Ab- and while there are some reduced connections between the MTL-AT with increased p-tau levels, increasing p-tau levels are much more robustly associated with reduced MTL-PM connectivity in CU Ab+.

f) Maass et al., report findings on functional task-based activity rather than functional connectivity

g) In the Introduction, the findings from Schröder et al., and Maass et al., are interpreted as indicating stronger associations between the MTL and the AT system. I believe these two papers report functional subregions in the entorhinal cortex that are differentially connected with two parallel processing streams (in the MTL and the neocortex) and the hippocampus, which means that the MTL is strongly connected with both the AT and PM systems.

h) The hypotheses in the Introduction could be formulated more clearly. What for example is meant by "tau pathology worsens"? It might be beneficial for the reader to refer to the stereotypical spatial progression pattern of tau accumulation.

i) In the fourth paragraph of the Introduction it should read, "An emerging study suggests that a functional connectome stability measure is associated with cognition (average and rate of episodic memory decline)."

Materials and methods and Results:

a) The rationale for the specific ROIs for MTL, AT, and PM networks is not clear, particularly as some of their selections differ from previous work. For example, the parahippocampal gyrus is a key region of the PM network, yet is placed in the MTL network here. Additionally, including more non-MTL AT regions, such as inferior temporal gyrus, could be beneficial to make the number of AT and PM regions more equivalent (3 regions/36 connections vs. 6 regions/60 connections), as more regions may inherently make the PM network more stable. Please cite appropriate references.

b) Could the authors include a visualization of the streamlines/connections between bilateral regions in order to understand whether that overlaps mainly with the aforementioned tracts. It would further help if the authors could elaborate in more detail on the underlying structural tracts, where they believe the signal is coming from and show examples of streamlines between MTL regions.

c) I was curious why the authors did not look at a pure baseline measure of memory performance at the first time point as a cleaner cross-sectional measure. Could the authors elaborate on that?

d) In the GLM, the authors report that their stability measure was associated with age (Model 2). However, they do not include age as a covariate in all other models. The same is true for other important covariates such as sex and years of education given that they are potentially related to CSF p-Tau and memory performance.

e) The authors should assess the relationships in models 3-5 while considering the effects of sex, age, and education. For model 5, given that stability is associated with age, how can the authors be sure that the moderating effect is driven by stability and not (also) by age? Regarding model 5, could the authors perform a mediation analysis to specifically test whether there is a significant indirect effect of CSF p-Tau via the stability measure on memory performance and decline while accounting for continuous age, sex, education and amyloid? In this framework the authors could also examine to what extent stability explains the effect of tau on memory (partial vs. full mediation).

f) "Stability" is interpreted as a positive aspect of structural integrity. However, it is possible that subjects with low MD measures at baseline stayed low at follow-up, and thus were "stable" (highly correlated) but have weak connectivity and low white matter integrity? This could be particularly common in the MTL in the MCI group, as damage from Ptau may have already occurred. Similarly, were there any subjects who had higher MD at follow-up, who were then "unstable" but increasing in connectivity (biologically less likely)? A breakdown of these different groups (high-MD stable, low-MD stable, high-low MD instable, low-high MD instable) and their prevalence in the CN/MCI samples would be informative.

g) Regarding the relationship between MTL structural stability, Ptau, and memory over 5 years in the whole sample – did you also test whether MTL-AT predicted average episodic memory? According to Figure 4, it appears that you did. Please include this info in the main text.

h) It would be helpful to have a figure (perhaps in the supplement) that shows the average mean diffusivity (MD) matrices for each group at each time point. This would allow readers to better understand the level of variability and raw values of these connectivity measures.

Discussion:

a) The finding of a stronger relationship between more Ptau and faster rate of memory decline in those with higher MTL stability (Figure 5) seems to run counter to the other findings where "stability" reflects structural integrity. Could this be explained by the "high stability" group containing both low-low MD and high-high MD subjects, allowing a strong relationship with cognition to emerge? I would like to see some consideration of this point in the Discussion.

b) In the stability group comparisons results (Figure 2A), the contrast between CN Ptau+ and MCI Ptau+ is currently not performed. From the figure, it appears that there may be a significant difference between those groups only in MTL-PM. This missing result would support their other findings because it would suggest that lower stability in MTL-PM connections might be related to the expression of clinical symptoms. Please amend and discuss accordingly.

c) While there was no interaction between CSF Aβ or neurodegeneration with Ptau on stability, was there a direct correlation between continuous levels of CSF Aβ or neurodegeneration with stability? If not, it helps support the finding that stability is specifically related to Ptau. Further, how do you interpret the age group interaction between Ptau and stability?

d) It is not clear why MTL-AT stability would not relate to episodic memory (average and rate of change) but MTL and MTL-PM would? Please discuss.

[Editors' note: further revisions were suggested prior to acceptance, as described below.]

Thank you for resubmitting your work entitled "Longitudinal stability of medial temporal lobe connectivity explains tau-related memory decline" for further consideration by *eLife*. Your revised article has been evaluated by Chris Baker as the Senior Editor and a Reviewing Editor.

The manuscript has been much improved but there is one remaining issue that needs to be addressed before acceptance, as outlined below:

The reviewers are very pleased with the manner in which you have addressed all of their concerns, however, they believe the Title warrants modification prior to publication. There was concern that the term "explains" is perhaps too strong given that there are many other factors that were not taken into account that might explain the effect demonstrated here. The reviewers have therefore suggested substituting "moderates" or "is associated with" rather than "explains".

Similarly, please ensure that any causal or overly strong claims are tempered in the Abstract and Discussion.

---

## [Author Response]

Essential revisions:1) Introduction:a) The authors state that hyperphosphorylated tau proteins are "necessary for subsequent Aβ aggregation". However, the prevailing view in the field is that Aβ aggregation is not dependent on concurrent tau pathology (ex. autosomal dominant AD). Please amend.

Thanks for pointing this out. We further clarified in the Introduction: “The dominant view in the literature is that Aβ pathology triggers tau aggregation, strongly supported by evidence from autosomal dominant AD (Hardy and Selkoe, 2002). However, there have been important recent findings from sporadic AD that suggest that hyperphosphorylated tau proteins that are no longer governed by normal cellular removal mechanisms play a necessary and underappreciated role in AD progression (Musiek and Holtzman, 2015), with some authors even suggesting this may be the primary step in the disease needed for subsequent AB aggregation (Arnsten et al., 2020, Braak and Del Tredici, 2015).”

b) The "PM" system is referred to as "posterior-MTL", but there are no MTL regions in their PM network. It would be helpful to use the terminology "posterior-medial" when referring to "PM", as well as citing Ranganath and Ritchey, 2012, which first introduced this framework.

The reviewer is completely correct. We have now used "posterior-medial" when referring to “PM” and cited Ranganath and Ritchey, 2012 in the Introduction.

d) In the Introduction the authors state that the "MTL is more strongly connected to the AT than PM systems", however this is not accurate. In the papers they cite (especially Navarro Schröder, 2015), some regions of the MTL (such as parahippocampal gyrus and posteromedial entorhinal cortex) are more strongly connected to the PM system.

The reviewer is correct. We further clarified in the introduction: “MTL is strongly connected with both the AT and PM systems, with anterolateral entorhinal cortex showing preferential connectivity with anterior regions overlapping with the AT system and parahippocampal gyrus and posteromedial entorhinal cortex with posterior regions overlapping with the PM system (Maass et al., 2015, Schröder et al., 2015)”.

e) I believe Berron et al., showed mostly reduced MTL-AT functional connectivity in CU Ab+ vs CU Ab- and while there are some reduced connections between the MTL-AT with increased p-tau levels, increasing p-tau levels are much more robustly associated with reduced MTL-PM connectivity in CU Ab+.

The reviewer is correct. To avoid confusion, we have taken this out from the Introduction.

f) Maass et al., report findings on functional task-based activity rather than functional connectivity.g) In the Introduction, the findings from Schröder et al., and Maass et al., are interpreted as indicating stronger associations between the MTL and the AT system. I believe these two papers report functional subregions in the entorhinal cortex that are differentially connected with two parallel processing streams (in the MTL and the neocortex) and the hippocampus, which means that the MTL is strongly connected with both the AT and PM systems.

We totally agree with review. We now modified the sentence in the Introduction:

“MTL is strongly connected with both the AT and PM systems, with anterolateral entorhinal cortex showing preferential connectivity with anterior regions overlapping with the AT system and parahippocampal gyrus and posteromedial entorhinal cortex with posterior regions overlapping with the PM system (Maass et al., 2015, Schröder et al., 2015).”

“Furthermore, tau pathology is related to worse performance in object memory tasks involving AT system relative to scene memory tasks involving PM system in cognitive normal older adults, matching the proposed spread of the disease via neuronal connections, as tau burden was much higher in the AT system compared with PM system in the early stages of AD (Arnsten et al., 2020, Maass et al., 2019).”

h) The hypotheses in the Introduction could be formulated more clearly. What for example is meant by "tau pathology worsens"? It might be beneficial for the reader to refer to the stereotypical spatial progression pattern of tau accumulation.

We further clarified in the Introduction:

“Second, given that tau tangles deposit early in the anterior temporal lobe (Braak and Del Tredici 2015, Maass et al., 2019), we hypothesized that Ptau first affects stability in the MTL and MTL-AT in the predementia stages, followed by MTL-PM as the disease progresses and tau pathology worsens and spreads to posterior brain regions”.

i) In the fourth paragraph of the Introduction it should read, "An emerging study suggests that a functional connectome stability measure is associated with cognition (average and rate of episodic memory decline)."

Thanks for pointing this out. We have corrected it.

Materials and methods and Results:a) The rationale for the specific ROIs for MTL, AT, and PM networks is not clear, particularly as some of their selections differ from previous work. For example, the parahippocampal gyrus is a key region of the PM network, yet is placed in the MTL network here. Additionally, including more non-MTL AT regions, such as inferior temporal gyrus, could be beneficial to make the number of AT and PM regions more equivalent (3 regions/36 connections vs. 6 regions/60 connections), as more regions may inherently make the PM network more stable. Please cite appropriate references.

This is a valid concern. Numerous studies have shown that medial temporal lobe (MTL) structures including the hippocampus and the surrounding hippocampal region consisting of the parahippocampal and entorhinal neocortical regions, are crucial for long-term memory. In terms of brain anatomy, parahippocampal gyrus receives input mainly from visuospatial association areas and provides major input to the entorhinal cortex, which in turn feeds directly into the hippocampus (Davachi et al., 2003). At the functional level, parahippocampal gyrus plays an important role in episodic memory relating to associative memory, source memory, and recollection (Aminoff et al., 2013). Given the anatomy and function of the parahippocampal gyrus, we consider it to belong to the MTL memory network.

We further clarified the rationale in the Results: “Since the hippocampus and the surrounding hippocampal region including the parahippocampal cortex and entorhinal cortex, are the primary regions supporting memory processing (Davachi et al., 2003), we consider them to belong to the MTL memory network. Thus, the MTL system includes entorhinal, hippocampus, and parahippocampal gyrus in both hemispheres.”

We included the bilateral inferior temporal gyrus into the non-MTL AT regions as the reviewer suggested and updated the results. We now added in the method: “In line with previous literature (Berron et a.,l 2020, Ranganath and Ritchey 2012), the AT system includes bilateral inferior temporal, temporal polar, lateral and medial orbitofrontal cortex, while the PM system includes bilateral posterior and isthmus cingulate, lateral occipital cortex, precuneus, and thalamus.”

b) Could the authors include a visualization of the streamlines/connections between bilateral regions in order to understand whether that overlaps mainly with the aforementioned tracts. It would further help if the authors could elaborate in more detail on the underlying structural tracts, where they believe the signal is coming from and show examples of streamlines between MTL regions.

Thanks for pointing this out. We have added the Figure 1B to visualize the streamlines between MTL related regions in a representative participant. We now added in the Results:

“For visualization purposes, we presented the connections generated between MTL related structures in a representative participant in Figure 1B. The connections within MTL include the hippocampal cingulum bundle and fornix. The connections between MTL and AT largely overlap with the anterior segments of the inferior longitudinal fasciculus (ILF) and inferior fronto-occipital fasciculus (IFOF), while the connections between MTL and PM mainly involve the cingulum bundle and posterior segments of ILF and IFOF.”

c) I was curious why the authors did not look at a pure baseline measure of memory performance at the first time point as a cleaner cross-sectional measure. Could the authors elaborate on that?

The stability measure is based on the information from both baseline and 2 year follow up. It’s more clinically meaningful to predict cognition in the future (up to four year follow-up) rather than the baseline collected prior to the stability measure. We now added in the Materials and methods:

“To assess the association between the longitudinal structural stability, baseline Ptau, and episodic memory, an average episodic memory score is calculated by averaging episodic memory over 5 years. Since we were interested in prediction of memory in the future, average memory score was used instead of baseline memory”.

d) In the GLM, the authors report that their stability measure was associated with age (Model 2). However, they do not include age as a covariate in all other models. The same is true for other important covariates such as sex and years of education given that they are potentially related to CSF p-Tau and memory performance.

Thanks for pointing this out. We have now examined relationships between the longitudinal structural stability, baseline Ptau, and episodic memory in the whole sample, including and excluding the covariates (age, education, sex, CSF Aβ, and neurodegeneration). Changes have been highlighted in yellow in the method and result sections.

e) The authors should assess the relationships in models 3-5 while considering the effects of sex, age, and education. For model 5, given that stability is associated with age, how can the authors be sure that the moderating effect is driven by stability and not (also) by age? Regarding model 5, could the authors perform a mediation analysis to specifically test whether there is a significant indirect effect of CSF p-Tau via the stability measure on memory performance and decline while accounting for continuous age, sex, education and amyloid? In this framework the authors could also examine to what extent stability explains the effect of tau on memory (partial vs. full mediation).

Thanks for pointing this out. We have now examined relationships between the longitudinal structural stability, baseline Ptau, and episodic memory in the whole sample, including and excluding the covariates (age, education, sex, CSF Aβ, and neurodegeneration). Changes have been highlighted in yellow in the method and result sections.

For model 5 (referred as model 6 in the revised manuscript), we now show that the moderating effect of MTL stability remained after the covariates were controlled (age, education, sex, CSF Aβ, and neurodegeneration).

We conducted the mediation analysis and didn’t find a significant indirect effect of CSF p-Tau via the stability measure on memory, or rate of memory change. Regardless of the lack of mediation effect, we did find a significant moderation effect of MTL stability on the relationship between tau and memory decline rate (see Model 6’ and Figure 6). To keep the paper more focused, we didn’t report these negative findings.

f) "Stability" is interpreted as a positive aspect of structural integrity. However, it is possible that subjects with low MD measures at baseline stayed low at follow-up, and thus were "stable" (highly correlated) but have weak connectivity and low white matter integrity? This could be particularly common in the MTL in the MCI group, as damage from Ptau may have already occurred. Similarly, were there any subjects who had higher MD at follow-up, who were then "unstable" but increasing in connectivity (biologically less likely)? A breakdown of these different groups (high-MD stable, low-MD stable, high-low MD instable, low-high MD instable) and their prevalence in the CN/MCI samples would be informative.

First, we want to clarify that the stability is a similarity measure, which reflects information in the patterns of MD across multiple connections, rather than only changes in mean MD. We agree with the reviewer that it’s possible that some subjects with low average MD at baseline and follow-up could show relatively high stability, while other could show low stability if their MD patterns changed across specific connections. We also want to clarify that, while it is possible that a loss of integrity causes poor stability due to an overall reduction in connections caused by pathology, it is also possible that pathology in certain connections could cause them to decrease connectivity, while other connections show corresponding increases as a result. Stability as a similarity measure is ambiguous to the mechanism by which pathology causes connections to change, but we believe that change in the overall pattern of connectivity in itself is meaningful.

To address the reviewer’s concern, we averaged the MD measures across all connections within MTL to get a mean MD score for MTL network. Then we used the median as a cutoff to dichotomize the mean MD score to “high” and “low”. Similarly, we dichotomized stability as “stable” vs. “instable”. The Table 2 in the manuscript displays the prevalence in the CN/MCI samples.

As we can see from Table 2, the prevalence of low-low stable individuals is actually not common in the MCI group (13.3%). In the Materials and methods, we now added:

“The traditional univariate analysis only considers the overall magnitude of the network (e.g., the average MD across all connections within MTL network). […] Thus, the traditional univariate analysis may not be as sensitive as the stability measure in detecting these subtle structural changes in the predementia AD stage.”

g) Regarding the relationship between MTL structural stability, Ptau, and memory over 5 years in the whole sample – did you also test whether MTL-AT predicted average episodic memory? According to Figure 4, it appears that you did. Please include this info in the main text.

Thanks for pointing this out. We did test whether MTL-AT predicted average episodic memory and we have now included this result in the main text and Figure 5 middle.

h) It would be helpful to have a figure (perhaps in the supplement) that shows the average mean diffusivity (MD) matrices for each group at each time point. This would allow readers to better understand the level of variability and raw values of these connectivity measures.

This is a good point. We have now added this figure in Figure 1—figure supplement 1.

Discussion:a) The finding of a stronger relationship between more Ptau and faster rate of memory decline in those with higher MTL stability (Figure 5) seems to run counter to the other findings where "stability" reflects structural integrity. Could this be explained by the "high stability" group containing both low-low MD and high-high MD subjects, allowing a strong relationship with cognition to emerge? I would like to see some consideration of this point in the Discussion.

This is a valid concern. As can be seen in Table 2, the stable group contains 12 high-high MD and 15 low-low MD subjects, while the instable group contains 12 high-high MD and 9 low-low MD subjects. As described above, since stability is calculated using all connections, rather than mean MD, instability is not restricted to high-low or low-high MD subjects. We now added in the Discussion:

“It is possible that the observation of a stronger relationship between more Ptau and faster rate of memory decline in those with higher MTL stability might be driven by greater variance in this group, caused by the inclusion of both stable individuals with overall low connectivity as well as consistently high connectivity. To assess this, we looked at the number of spread of individuals across these groups based on their mean structural connectivity (Table 2). The stable group contains 12 high-high MD and 15 low-low MD subjects, while the instable group contains 12 high-high MD and 9 low-low MD subjects. While there are more low-low subjects in the stable group, the ratio of low-low to high-high is similar across these groups, suggesting increased variance is unlikely to drive the strong relationship with cognition in the high stability group.”

b) In the stability group comparisons results (Figure 2A), the contrast between CN Ptau+ and MCI Ptau+ is currently not performed. From the figure, it appears that there may be a significant difference between those groups only in MTL-PM. This missing result would support their other findings because it would suggest that lower stability in MTL-PM connections might be related to the expression of clinical symptoms. Please amend and discuss accordingly.

This is an excellent point. We have conducted the contrast between CN Ptau+ and MCI Ptau+ and updated the results and discussion accordingly.

In the Results, we modified the sentences: “In line with our hypothesis that MTL and MTL-AT stability is affected by Ptau early in the predementia AD stage, compared to CN Ptau-, structural stability was significantly lower in CN Ptau+ in MTL (t(42)=-2.11, p = 0.022) and MTL-AT (t(42)=-2.57, p = 0.007). Decreased structural stability was further found in MTL (t(54)=-2.61, p = 0.006) and MTL-AT (t(54)=-3.55, p < 0.001) in MCI Ptau+, compared to CN Ptau-. In line with our hypothesis that MTL-PM is affected later in disease progression, we found a significantly lower stability in MTL-PM in MCI Ptau+, compared to CN Ptau+ (t(46)=-5.12, p < 0.001) and CN Ptau- (t(54)=-3.89, p < 0.001, Figure 2A).”

In Discussion, we modified the sentence: “CN Ptau+ and MCI Ptau+ had decreased stability in MTL and MTL-AT compared to CN Ptau-, while MCI Ptau+ further showed decrease in MTL-PM stability relative to both CN Ptau- and CN Ptau-+ groups, suggesting that Ptau relates to structural stability in a temporal pattern that matches its proposed spread from the MTL via neuronal connections”.

c) While there was no interaction between CSF Aβ or neurodegeneration with Ptau on stability, was there a direct correlation between continuous levels of CSF Aβ or neurodegeneration with stability? If not, it helps support the finding that stability is specifically related to Ptau. Further, how do you interpret the age group interaction between Ptau and stability?

We have now examined the direct correlation between continuous levels of age, CSF Aβ or neurodegeneration with stability in model 2 and updated the result in the main text and Figure 3. In the Results, we added now:

“We examined whether structural stability could be directly affected by covariates (i.e., age, CSF Aβ or neurodegeneration) in the entire sample with Model 2 (Y *stability* = β20 + β21*Covariates* + ԑ_2_). We found no significant relationship between age and stability (all ps > 0.05, Figure 3A). We found a significant correlation between CSF Aβ and stability in MTL-PM (β = 0.40, SE = 0.16, Wald’s χ2 = 6.04, BH-adjusted p = 0.042, Figure 3B). Neurodegeneration significantly related to stability within MTL (β = 0.58, SE = 0.26, Wald’s χ^2^ = 4.97, BH-adjusted p = 0.039), MTL-AT (β = 0.49, SE = 0.25, Wald’s χ^2^ = 3.89, BH-adjusted p = 0.048) and MTL-PM (β = 0.61, SE = 0.19, Wald’s χ^2^ = 10.39, BH-adjusted p = 0.003, Figure 3C).”

Regarding the age group interaction between Ptau and stability, after we included the inferior temporal cortex into AT regions, there was no significant age group interaction (Figure 4A).

d) It is not clear why MTL-AT stability would not relate to episodic memory (average and rate of change) but MTL and MTL-PM would? Please discuss.

After we included the inferior temporal cortex into the non-MTL AT regions as one reviewer suggested, MTL-AT stability predicted episodic memory (average episodic memory). We have updated the Results in the main text and Figure 5.

[Editors' note: further revisions were suggested prior to acceptance, as described below.]

The manuscript has been much improved but there is one remaining issue that needs to be addressed before acceptance, as outlined below:The reviewers are very pleased with the manner in which you have addressed all of their concerns, however, they believe the Title warrants modification prior to publication. There was concern that the term "explains" is perhaps too strong given that there are many other factors that were not taken into account that might explain the effect demonstrated here. The reviewers have therefore suggested substituting "moderates" or "is associated with" rather than "explains".

We totally agree. We now used "is associated with" instead of “explains”. The current title is “Longitudinal stability of medial temporal lobe connectivity is associated with tau-related memory decline**”.**

Similarly, please ensure that any causal or overly strong claims are tempered in the Abstract and Discussion.

We have altered the claims and made them less causal in the abstract, introduction and discussion:

In the Abstract, we wrote: “…this loss of stability in connections known to be important for memory moderated the relationship between Ptau accumulation and memory decline.”

In the Introduction, we wrote: “Finally, given that MTL and related structures play a central role in episodic memory, we hypothesized that MTL related structural stability might be associated with tau-related episodic memory decline.”

In the Discussion, we wrote: “… whether this loss of stability in connections important for episodic memory is related to the link between Ptau accumulation and cognitive decline.”

“Importantly, structural stability in MTL moderated the effect of Ptau on the rate of memory change, suggesting that structural stability in MTL may be a candidate as a mechanistic link between the accumulation of Ptau and cognitive decline.”

“A model showed that decreases in the stability of within MTL connections significantly predicted episodic memory decline, suggesting changes in these connections resulting from increased Ptau is associated with decreases in episodic memory.”